# A social chemosignaling function for human handshaking

Idan Frumin[1]*, Ofer Perl[1], Yaara Endevelt-Shapira[1], Ami Eisen[1], Neetai Eshel[1], Iris Heller[1], Maya Shemesh[1], Aharon Ravia[1], Lee Sela[1], Anat Arzi[1], Noam Sobel[1]*

[1]Department of Neurobiology, Weizmann Institute of Science, Rehovot, Israel

**Abstract** Social chemosignaling is a part of human behavior, but how chemosignals transfer from one individual to another is unknown. In turn, humans greet each other with handshakes, but the functional antecedents of this behavior remain unclear. To ask whether handshakes are used to sample conspecific social chemosignals, we covertly filmed 271 subjects within a structured greeting event either with or without a handshake. We found that humans often sniff their own hands, and selectively increase this behavior after handshake. After handshakes within gender, subjects increased sniffing of their own right shaking hand by more than 100%. In contrast, after handshakes across gender, subjects increased sniffing of their own left non-shaking hand by more than 100%. Tainting participants with unnoticed odors significantly altered the effects, thus verifying their olfactory nature. Thus, handshaking may functionally serve active yet subliminal social chemosignaling, which likely plays a large role in ongoing human behavior.

## Introduction

Social chemosignaling plays a large role in mammalian and particularly rodent behavior (*Dulac and Torello, 2003*; *Keverne, 2005*; *Stowers and Marton, 2005*; *Brennan and Zufall, 2006*; *Kaur et al., 2014*), and therefore mammals typically greet each other with careful olfactory investigation (*Doty, 1986*; *Drea et al., 2002*; *Wesson, 2013*). Similarly, there is mounting evidence for the role of social chemosignaling in ongoing human behavior (*McClintock, 2000*; *Wysocki and Preti, 2004*; *de Groot et al., 2012*). Human social chemosignals drive menstrual synchrony (*Stern and McClintock, 1998*), serve in mate selection (*Jacob et al., 2002*), convey fear (*Chen et al., 2006*; *Zhou and Chen, 2009*), drive pronounced hormonal (*Preti et al., 2003*; *Wyart et al., 2007*; *Gelstein et al., 2011*) and behavioral (*Jacob et al., 2001a*; *Bensafi et al., 2003*) modifications, and alter brain activity (*Sobel et al., 1999*; *Jacob et al., 2001b*; *Savic et al., 2001, 2005*; *Lundström et al., 2006*). Although there remains controversy on the statistics of menstrual synchrony (*Schank, 2002*), and on application of the term *pheromone* to instances of social chemosignaling in humans (*Meredith, 2001*; *Wysocki and Preti, 2004*; *Doty, 2010*), that humans emit odors that can influence behavior and perception in other humans is largely agreed upon. Unlike other mammals, however, humans do not engage in overt olfactory sampling and investigation of conspecifics. Thus, how do humans obtain the social chemosignals they so clearly process? Although some human cultures include explicit olfactory sampling in stereotypical greeting behaviors (*Classen, 1992*), and common behaviors such as hugging and kissing provide ample opportunity for covert olfactory sampling (*Nicholson, 1984*), human overt olfactory sampling and investigation of unfamiliar individuals is largely a taboo. Here, we asked whether human handshaking might serve as a subliminal mechanism for sampling social chemosignals. Handshaking is common across cultures and history (*Firth, 1972*; *Schiffrin, 1974*), yet its functional antecedents remain unclear, and the commonly cited notion of gesturing no weapons in the shaking hand has only limited scholarly support.

*For correspondence: idan.frumin@gmail.com (IF); noam.sobel@weizmann.ac.il (NS)

Competing interests: The authors declare that no competing interests exist.

**eLife digest** Animals often sniff each other as a form of greeting to communicate with each other through chemical signals in their body odors. However, in humans this form of behavior is considered taboo, especially between strangers.

Scientists argue that, in spite of our efforts to avoid being 'smelly', we may actually smell each other without being aware that we do so. Here, Frumin et al. first put on latex gloves and then shook hands with volunteers to collect samples of their odor. Chemical analysis of the gloves found that a handshake alone was sufficient to transfer the volunteers' odor. These odors were made of chemicals that are similar to ones that animals smell when sniffing each other.

Therefore, when we shake hands with a stranger, it is possible that we may inadvertently smell the stranger's chemical signals. To address this possibility, Frumin et al. investigated how humans behave after shaking hands with a stranger. Volunteers were asked to wait in a room alone before they were greeted by one of the researchers. Some of these volunteers were greeted with a handshake and others were greeted without a handshake. Afterwards, all the volunteers spent some time in a room by themselves where their behavior was covertly monitored.

Frumin et al. found that volunteers who shook hands were more likely to sniff their hand, for example, by touching their nose when they were in the room on their own, than those who did not shake hands. After the volunteers shook hands with someone of their own gender, they spent more time sniffing their right hand (the one they had used for the handshake). However, after the volunteers shook hands with someone of the opposite gender, they spent more time sniffing their left hand instead.

Next, the body odor of some of the experimenters was tainted by perfumes or gender-specific odors. Volunteers who shook hands with these tainted individuals behaved differently; when the experimenter was tainted with perfume the volunteers spent more time sniffing their own hands, but when the experimenter was tainted with a gender-specific odor they spent less time sniffing of their own hands. This shows that different smells influenced the hand sniffing behavior of the volunteers.

Frumin et al.'s findings suggest that a simple handshake may help us to detect chemical signals from other people. Depending on the person's gender, we may respond by sniffing our right hand to check out the person's odor, or our left hand to smell ourselves in comparison. Future studies will involve finding out how this sniffing behavior could work as an unconcious form of human communication.

## Results

### Handshakes can transfer relevant skin-bound molecules

Handshakes are sufficient for the transfer of various pathogens (*Mela and Whitworth, 2014*), and it is therefore likely that they are sufficient for the transfer of chemosignals as well. To test whether the general type of molecules implicated in chemosignaling can also be transferred by handshake, we used gas-chromatography mass-spectrometry to sample surgical nitrile gloves before and after a handshake with the bare hand of 10 individuals (5F, mean age = $34.1 \pm 5.6$ years) (*Figure 1A*) (see 'Materials and methods'). Examination of the resulting chromatograms (*Figure 1B*) revealed several peaks that were all effectively transferred through handshake alone. These included previously identified compounds of interest in human bodily secretions (*Gallagher et al., 2008*), such as squalene, which is a putative social chemosignaling component in several species including dogs (*Apps et al., 2012*) and rats (*Achiraman et al., 2011*); hexadecanoic acid, which is a putative social chemosignaling component in both mammals (*Briand et al., 2004*) and insects (*Tang et al., 1989*); and geranyl acetone, which is present in human secretions (*Gallagher et al., 2008*), but to date was considered a social chemosignaling component in insects alone (*Zarbin et al., 2013*) (*Figure 1C*). Each of these three compounds was transferred by handshake in all 10 of 10 subjects but not once in the control (all $t[9] > 3.9$, all $p < 0.003$). Use of cosmetics beyond hand-soap by these 10 subjects was minimal (2 subjects), and there was overlap in only one brand of hand-soap. We therefore submitted this hand-soap to GCMS analysis as well, and did not detect any traces of the three above components. Thus, we conclude that these were likely endogenous skin-bound molecules. These

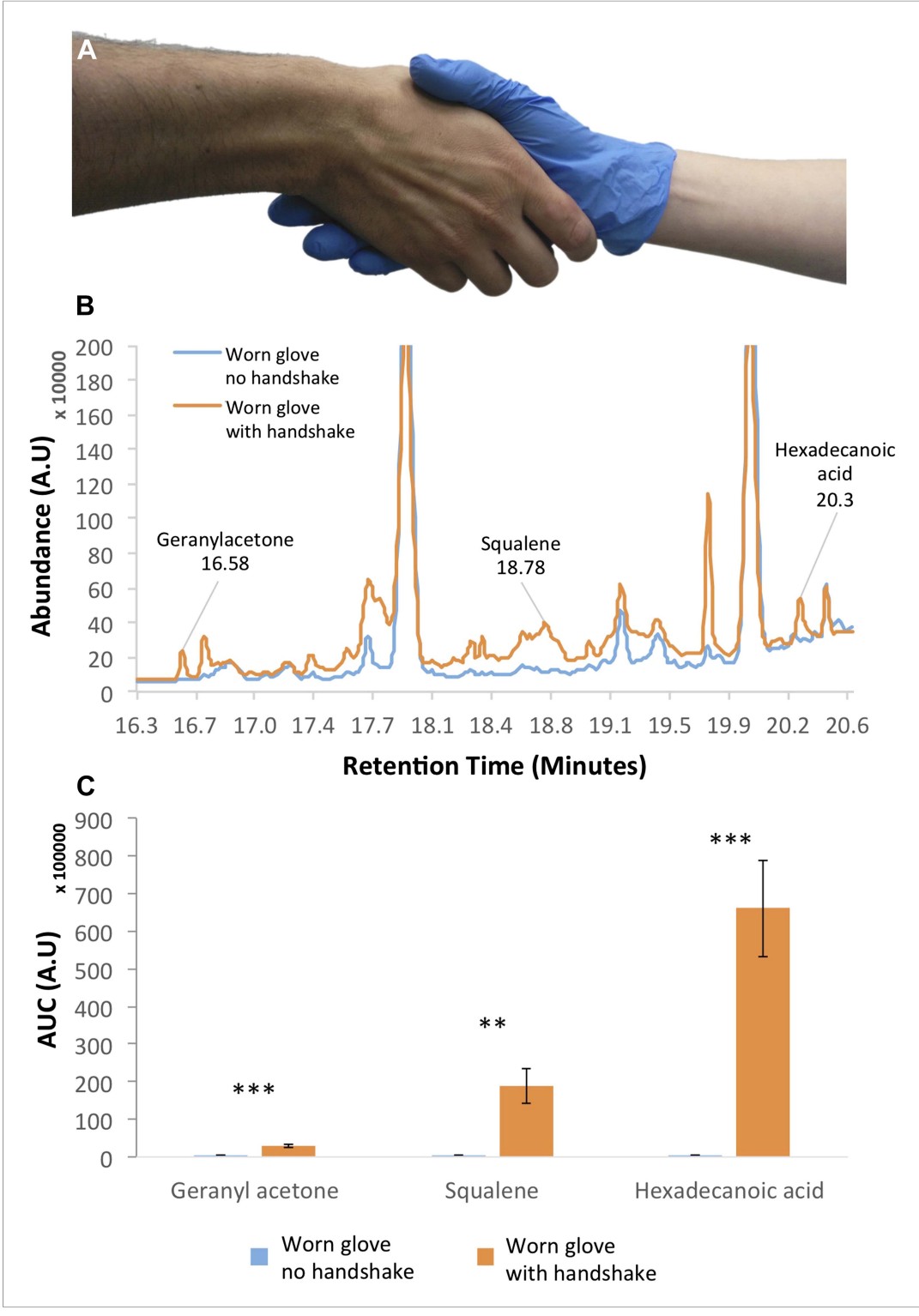

**Figure 1**. Handshakes can transfer chemosignaling components. (**A**) A representative image of our sampling method using a nitirle glove during handshake. (**B**) An example chromatogram from one experiment. Note that the 'clean' condition is a glove worn by the same hand, but not shaken. This controls for potential contamination from the glove-wearing hand. Most unmarked peaks in the chromatogram that are present in both the clean and the shaken are PDMS artifacts—various siloxane compounds that come from either the twister used to sample the gloves or the GC column. Moreover, some peaks that are present in this example were not present across subjects. The only three peaks that were present following all shakes but never once in control are those we describe in the

*Figure 1. continued on next page*

*Figure 1. Continued*
following panel. (**C**) Summated data from 10 individuals (each an average of three shakes) demonstrating three compounds of interest in chemosignaling (*Gallagher et al., 2008*) that were effectively transferred by handshaking in all instances and never once in control. Error bars are standard error, **p < 0.01, ***p < 0.001.

results do not imply that these molecules are necessarily social chemosignaling components in humans, but they do demonstrate that the act of handshaking is sufficient to transfer molecules of the type that are likely relevant to mammalian social chemosignaling.

## Humans often sniff their own hands

To test whether humans use handshakes to sample conspecific chemosignals, we devised a structured paradigm. Subjects who were invited to our lab for participation in experiments were first led to a room where they were requested to sit and wait. About 3 min later a cosmetics-free experimenter entered the room, introduced him/her-self using a fixed greeting text (20 ± 8 s duration) either with or without a handshake and ended in telling the subject that they would soon return to start the experiment. These ~20 s are referred to from hereon as the 'greet'. The subject was then again left alone in the room for an additional 3 min. The entire interaction was filmed with hidden cameras. Because human chemosignaling is influenced by gender (*Savic et al., 2001*; *Bensafi et al., 2003*; *Radulescu and Mujica-Parodi, 2013*) (F/M), we aimed for ~20 subjects per each possible experimenter (exp) to subject (sub) gender interaction (Fexp/Fsub; Fexp/Msub; Mexp/Msub; Mexp/Fsub), and further interleaved experiments once with handshake and once without (baseline control), culminating in an intended ~160 subjects for analysis. We therefore recruited 175 subjects into this paradigm (84F, mean age = 26.49 ± 3.69 years), who each shook hands with one of 20 different experimenters (13F, mean age = 35.24 ± 6.38 years).

The film data were then scored for potential olfactory hand sampling behavior. Criterion for scoring was any application of a hand to the face, as long as touching was under the eyebrows and above the chin. Left (non-shaking) and right (shaking) hands were scored separately. Although these scoring criteria are largely unequivocal, two researchers independently scored the data, and we then tested for inter-rater agreement. The correlation between raters regarding duration of face-touches was r = 0.96, p < 0.0001 (*Figure 2A*), implying that scoring the data using these strict criteria was largely uninfluenced by rater. Through this process, we also omitted 22 subjects from further analysis due to non-compliance (typically using a cell-phone during the experiment), retaining 153 subjects, 80 who experienced greets with handshake, and 73 who experienced greets without handshake. All the data from these subjects are available in *Supplementary file 1*, sheet 1.

Next, for each subject we summed the time each hand spent at the vicinity of the nose (i.e., under the eyebrows and above the chin only) across 1 min before (+greet event time) and 1 min after (+ greet event time) the greeting event (which culminated at ~80 ± 16 s given the added time of the greet event itself, see *Figure 2—figure supplement 1*). Consistent with previous studies (*Nicas and Best, 2008*), we observed that humans often bring their hands to their noses (see online *Video 1*). Of 153 subjects, 85 (55.55%) touched their nose with their hand at least once during baseline before the greet. The average time of a hand at the nose across these ~80 s was 5.38 ± 15.7 s for the right hand, and 12.33 ± 23.81 s for the left hand. An analysis of variance (ANOVA) revealed that this left over right hand preference was significant (F[1,151] = 8.14, p = 0.005), and there was no difference across genders in this behavior (F[1,151] = 0.3, p = 0.86) (*Figure 2B*). Combined, this amounts to ~17 s, in other words idle subjects had a hand (either right or left) at the vicinity of their nose for 22.14% of the time. To explore the spatial properties of this behavior, we parsed each face into 17 regions (*Figure 2C*) and coded the region-specific touching. This representation revealed that facial touching was mostly in regions at or under the nose, rendering the touching hand potentially well placed for olfactory exploration (*Figure 2C*).

Whereas facial self-touching has been considered a form of displacement stress response (*Troisi, 2002*), akin to rodent grooming, the novel framework we propose here for this behavior is that of olfactory sampling. Although we think that the video data are strongly supportive of this view (see online *Video 1*), to further estimate whether bringing the hand to the nose is associated with olfactory

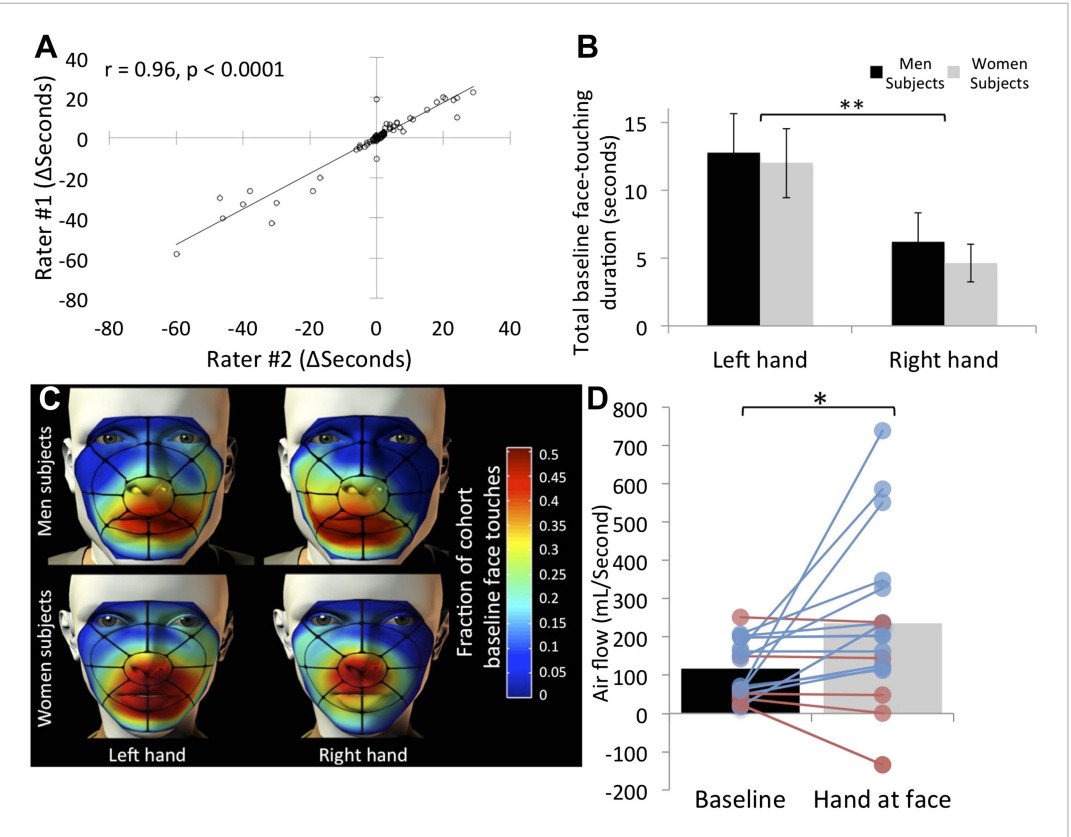

**Figure 2**. Humans often touch their own face and concurrently sniff. (**A**) Agreement in scoring of 153 subjects across two independent raters. (**B**) Total face touching duration during the 1-min baseline. (**C**) Spatial distribution of face touching during the 1-min baseline. Grid reflects 17 facial regions. (**D**) Measure of nasal airflow during baseline vs the time when a hand was at the face. Subjects that increased flow in blue and subjects that decreased flow in red. Solid bars reflect the mean. Error bars in **B** are standard error. **p < 0.01. *p < 0.05.

The following figure supplement is available for figure 2:

**Figure supplement 1**. Experimental time-course.

exploration, we repeated the task in 33 additional subjects (26F, mean age = 23.84 ± 5.36 years) with concurrent measurement of nasal airflow. To measure nasal airflow, we fitted subjects with a nasal cannula (*Johnson et al., 2006*). To avert subject attention from any interest in nasal airflow, we also fitted them with several mock psychophysiology electrodes (e.g., ECG), and told them they were participating in an electrophysiology rig equipment calibration and testing procedure. Such tethered subjects behaved differently, reducing the prevalence of hand exploration from the previously observed ~22% of the time to ~11% of the time. Nevertheless, this generated a sufficient number of events for analysis (17 hand sampling subjects). We found that when a hand was at the vicinity of the nose, nasal airflow more than doubled over baseline (baseline flow = 112.75 ± 75.56 ml/s, hand-at-face flow = 237.81 ± 220.82 ml/s, t[16] = 2.37, p = 0.03) (*Figure 2D*) (online *Video 2*). In other words, when subjects brought their hand to their nose, they concurrently sniffed.

## Increased hand investigation after handshakes within gender

Having found that handshakes are sufficient to transfer molecular components of the type typically involved in social chemosignaling, and that humans often bring their hands to their nose and sniff, we next set out to directly test our hypothesis that handshaking subserves social chemosignaling. We first computed for each hand a change score reflecting the time spent at the nose across ~80 s after the greet minus the time spent at the nose across ~80 s before the greet (*Supplementary file 1*, sheet 1).

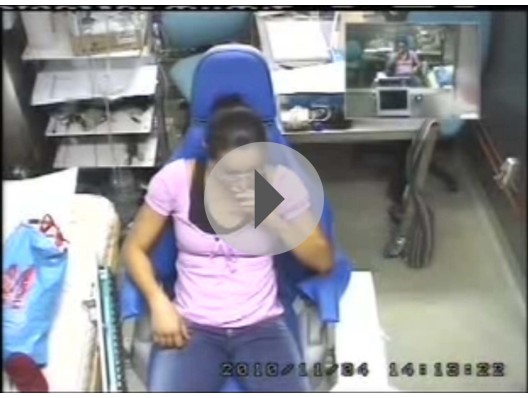

**Video 1.** Humans often sniff their own hands. This is an assortment of scored events from across the data (before, during, and after greet), demonstrating that humans often apparently sniff their own hands. Note that these are not the 'best cases', as typically subjects who engaged in very overt self-sampling did not later consent to use of their video in publication. DOI: 10.7554/eLife.05154.006

In other words, positive values indicate an increase in hand exploration after the greet. Using change scores accounts for any individual differences in face touching. Next, from each change score, we subtracted the mean of the no-handshake control for that specific interaction (either Fexp/Fsub; Fexp/Msub; Mexp/Msub; Mexp/Fsub), such that for each subject we now have a change from condition-specific baseline (*Supplementary file 1*, Sheet 3) (note that we also replicate the analysis without this step, see *Figure 3—figure supplement 1*). We then conducted a repeated measures omnibus ANOVA with factors of subject gender (M/F), experimenter gender (M/F), and a dependent repeated compact variable of exploration change time for right (shaking) and left (non-shaking) hands (hand). Moreover, we concurrently analyzed the data for each hand separately using non-parametric tests as well (see comment on statistics in 'Materials and methods').

In brief, this analysis primarily revealed that both men and women significantly increased exploration of the hand that shook after shaking hands within gender. By contrast, after shaking hands across gender, both men and women decreased right (shaking) hand exploration to a level below baseline, yet increased exploration of the left (non-shaking) hand to significantly above baseline. In more detail:

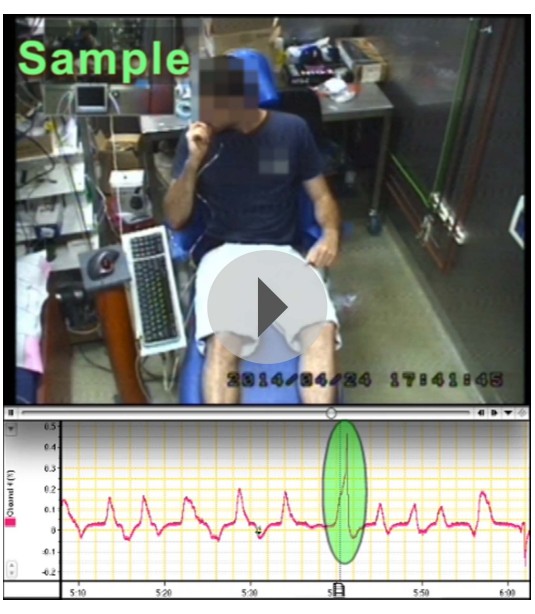

**Video 2.** Hand sampling is concurrent with sniffing. The video is from the control experiment that included a nasal cannula for nasal airflow recording. The airflow cursor is time-locked with the video. Note that when the subject was scored as sampling, he concurrently sniffed. A second example is in *Video 4*. This individual is obscured by pixalization to reflect requested level of privacy. DOI: 10.7554/eLife.05154.007

The ANOVA revealed a main effect of experimenter gender ($F[1,77] = 7.28$, $p = 0.009$) and an overwhelming three-way interaction ($F[1,77] = 37.79$, $p < 0.0001$) (see comments in final paragraph of 'Materials and methods'). The main effect reflected that in both men and women equally ($F[1,77] = 0.19$, $p = 0.66$), handshakes from male experimenters elicited increased ensuing sampling of the left non-shaking hand (mean Fexp = $-0.82 \pm 17.55$ s, mean Mexp = $7.54 \pm 14.95$, $t[79] = 2.29$, $p = 0.025$, non-parametric reanalysis: Mann–Whitney U, $Z = 3.17$, $p = 0.001$), and a trend in this direction for the right shaking hand (we refer to this as a trend because the effect was not evident in the parametric analysis, but was evident in the non-parametric approach: mean Fexp = $-1.18 \pm 19.06$ s, mean Mexp = $2.81 \pm 11.55$, $t[79] = 1.12$, $p = 0.26$, non-parametric reanalysis: Mann–Whitney U, $Z = 1.98$, $p = 0.048$).

The three-way interaction reflected that for both men and women equally ($F[1,77] = 0.18$, $p = 0.67$), exploration of the right shaking hand increased after shaking the hand of an individual from the same gender, yet decreased after shaking the hand of an individual from the opposite gender (within gender = $7.34 \pm 8.16$ s, across gender = $-5.79 \pm 18.99$ s, $t[79] = 4.02$, $p = 0.0001$, non-parametric reanalysis: Mann–Whitney U, $Z = 6.05$,

p < 0.0001) (*Figure 3A,B*) (online *Video 3*) (*Supplementary file 1*, sheet 4). In other words, individuals significantly increased right hand exploration following same gender greets that contained a handshake. These subjects (within gender with handshake) shifted in right shaking hand sampling from an expected $-5.39 \pm 15.29$ s (expected = change following no-handshake greet) to $2.14 \pm 8.1$ s (change following handshake greet), that is, a 135.99% increase. In contrast, sampling of the left non-shaking hand decreased after shaking the hand of an individual from the same gender, yet increased after shaking the hand of an individual from the opposite gender (within gender = $-4.86 \pm 17.5$ s, across gender = $10.87 \pm 11.89$ s, $t[79] = 4.74$, $p < 0.0001$, non-parametric reanalysis: Mann–Whitney U, $Z = 5.62$, $p < 0.0001$) (*Figure 3A*). These subjects (across gender with handshake) shifted in left non-shaking hand sampling from an expected $-7.95 \pm 28.33$ s (expected = change following no-handshake greet) to $2.91 \pm 11.84$ s (change following handshake greet), that is, a 139.24% increase. Taken together, these data imply that after shaking hands with individuals from across gender humans increase left non-shaking hand sampling, yet after shaking hands with individuals of the same gender humans robustly selectively increase sampling of the hand that shook (see online *Video 4*). Note that replicating the analysis without correcting for condition baseline generated the same outcome (*Figure 3—figure supplement 1*). Again, to estimate whether these were touches allowing olfactory sampling, we analyzed the spatial distribution of touch. We found that the right hand increase in touching following within gender handshakes was directly at the vicinity of the nose (*Figure 3C*). To further characterize this behavior, we also analyzed the latency to hand exploration. The average latency following greet onset in those subjects that explored the shaking hand was $25.65 \pm 16.48$ s (*Figure 3D*). In order to include all subjects in the latency analysis (i.e., not only those that face-touched that are in *Figure 3D*), we are forced to assign an arbitrary latency of 60 s + greet time ($20 \pm 8$) to subjects who never self-explored. Moreover, latency lacks a subject-specific baseline because we do not have a baseline event from which to measure it, so we can only calculate a condition-specific baseline. Finally, 14 subjects had their left non-shaking hand continuously at their nose before, during, and after the greet, and therefore we should not calculate latency for the left hand in these subjects. With these limitations in mind, we further analyzed the right shaking hand only. An ANOVA on latency change scores revealed a trend towards a main effect of experimenter gender ($F[1,57] = 3.57$, $p = 0.06$) reflecting that women experimenters trended towards eliciting faster sampling responses regardless of subject gender (change from condition-specific baseline, women experimenters = $-14.27 \pm 29.72$ s, men experimenters = $0.36 \pm 29.43$ s, $t[59] = 1.92$, $p < 0.06$). The analysis of latency revealed no significant effects beyond this trend.

Given that 20 experimenters alternated in the role of handshakers, it is unlikely that a particular individual drove these results. Nevertheless, because some experimenters shook significantly more hands than others, we conducted an ANOVA with a single factor of experimenter and a dependent variable of exploration change time, and found no effect for the left ($F[19,133] = 0.46$, $p = 0.97$) or right ($F[19,133] = 0.54$, $p = 0.94$) hands. Moreover, individual comparisons revealed that although two experimenters drove more ensuing exploration than others (both $p < 0.05$), these differences did not survive correction for multiple comparisons. Thus, although we speculate that some individuals may drive such effects more than others, the design of our study largely protected against such influence in the current results. A second potential source of individual variance is subject handedness. Consistent with population distribution, 15 of the 153 subjects and two of the 20 experimenters were left-handed. This retained four left-handed subjects in the 'with handshake within gender' condition, three of which (75%) increased investigation of the shaking hand (right) after handshake. This reflects a trend towards a stronger effect in left-handed subjects, but this difference is not statistically different from the remaining right-handed subjects ($X^2 = 2.12$, $p = 0.14$). Thus, here too our design protected against influence of individual differences such as handedness, and we cannot say whether handedness impacts this behavior.

## Tainting experimenters with odors altered the effect

Given that subjects increased sampling of both their right hand that shook (within gender greets) and their left hand that did not shake (across gender greets), one may suggest that the latter effect calls into question the olfactory sampling nature of the behavior we observed. Despite the location of touching at the nose (*Figure 2C*), and the pronounced concurrent sniffing (*Figure 2D*), perhaps this remains a form of non-olfactory displacement stress response (*Troisi, 2002*). On this front, we first

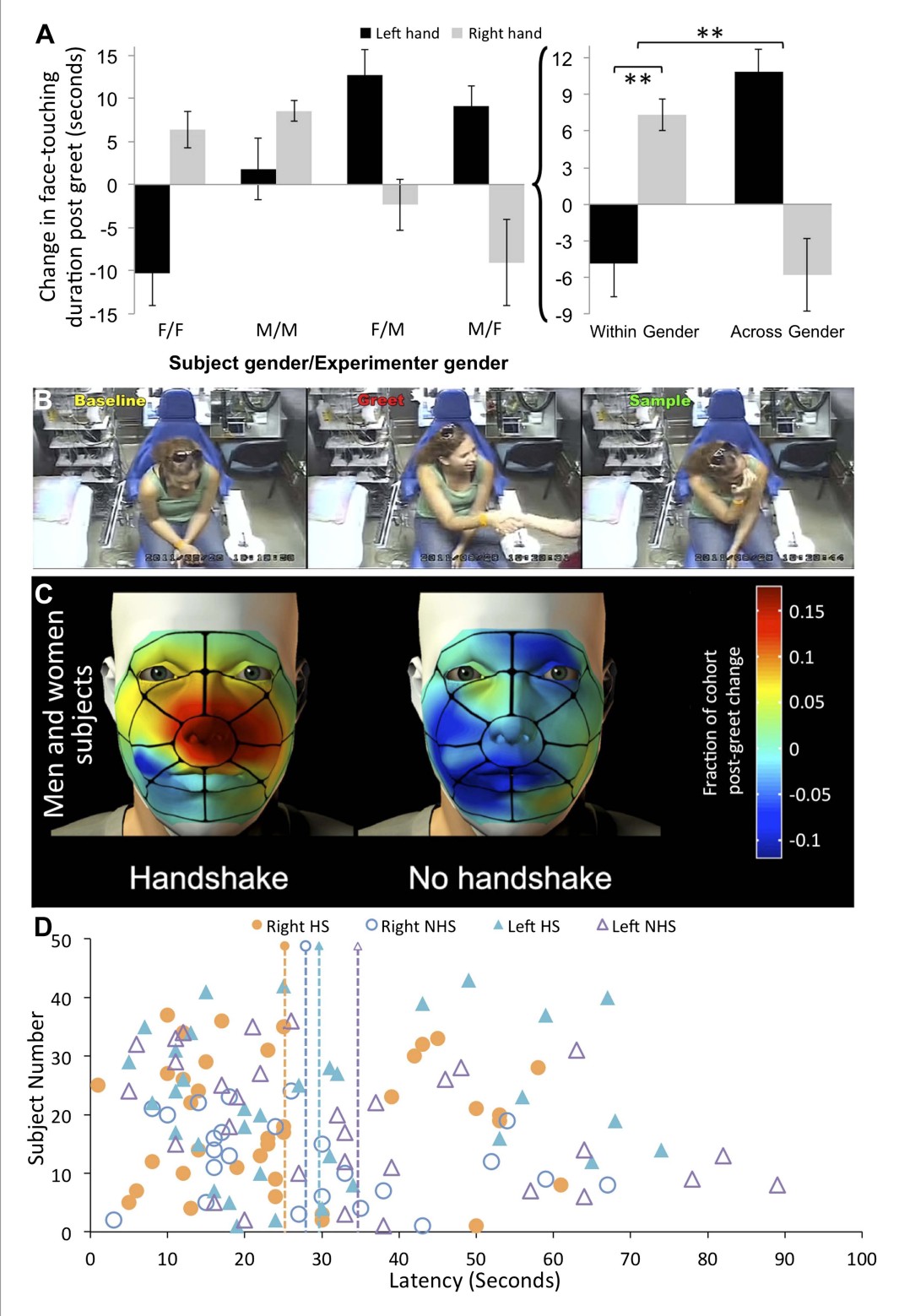

**Figure 3**. Humans sniff their own hands after handshake. (**A**) Right and left hand changes in duration of face-touching following a greet. Duration change scores are after individual-baseline and condition-baseline normalization. The lettering under each pair of columns (e.g., F/F) reflects the 'Subject gender/Experimenter gender' interaction, respectively. The summation on the right is the interaction reflecting increased sampling of the right hand following within gender greets with handshakes, and increased sampling of the left hand following cross-

*Figure 3. Continued*

gender greets without a handshake. (**B**) Three screen-shots depicting from left to right: a subject during baseline before the greet, then during handshake greet, and finally self-sampling after the experimenter leaves the room (see *Video 3*). (**C**) The spatial distribution of change in right-handed face-touching following the greet. (**D**) Latency to face-touch in the handshake (HS) and no-handshake (NHS) conditions. The figure contains only subjects who touched their face within the analysis time window. The 14 subjects with left hand continuously at face before during and after the greet were omitted from the figure. The dotted lines reflect the mean for each condition. Error bars are standard error. \*\*p < 0.01.
The following figure supplement is available for figure 3:

**Figure supplement 1**. Reanalysis without correcting for condition-specific baseline.

---

must stress that much of the behavior we scored was after the experimenter left the room (online *Video 3*). As noted above, mean greet duration was 20 ± 8 s, and mean sample latency was 25.65 ± 16.48 s. In other words, subjects were often alone in the room when they engaged in the measured behavior, and therefore this was mostly not a direct concurrent response to the presence of the experimenter. Nevertheless, we set out to conduct an additional control experiment.

To further investigate the olfactory nature of the observed effects, we again measured the behavior, yet here unbeknownst to the subjects we tainted the experimenters with odors (*Figure 4A*). To maintain a manageable scope, we now limited our effort to the 'within gender with handshake' condition in women alone. We added three experimental groups: one where women experimenters were tainted with the putative male social chemosignal 4,16-androstadien-3-one (AND) (*Savic et al., 2001*; *Huoviala and Rantala, 2013*) (n = 22), one where women experimenters were tainted with the putative female social chemosignal estra-1,3,5(10),16-tetraen-3-ol (EST) (*Savic et al., 2001*; *Huoviala and Rantala, 2013*) (n = 20), and one where women experimenters were tainted with a commercial unisex perfume (CK-be) (n = 21). We compared these data to the

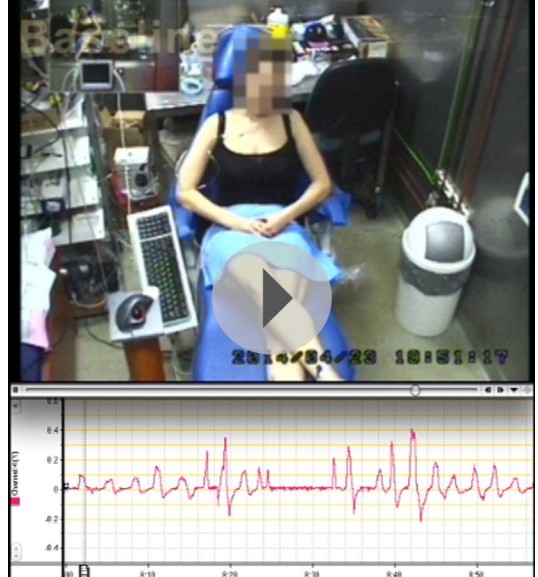

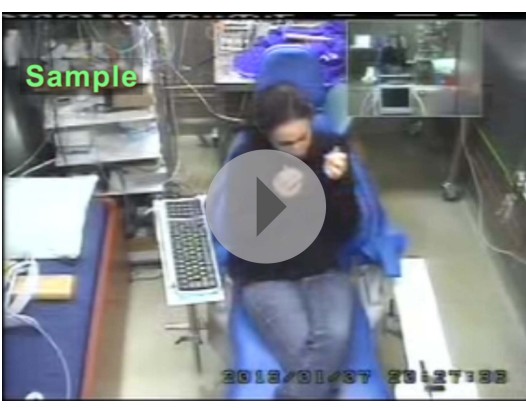

**Video 3.** Humans sample the hand that shook. Several greet events with ensuing behavior. The text in the upper left corner denotes the scored condition. Again, these are far from 'best cases', as typically subjects who engaged in very overt self-sampling did not later consent to use of their video in publication. Finally, some individuals are obscured by pixalization to reflect requested level of privacy.

**Video 4.** Pronounced sniffing of the hand that shook. An example from the control experiment that included a nasal cannula for nasal airflow recording. Although this may seem like a staged dramatization, it is not. This is raw data, with an explicit self-sample that occurred the moment the experimenter ended the greet and left the room. As the frozen image at the end highlights, this self-sampling behavior was perfectly timed with a pro-nounced sniff.

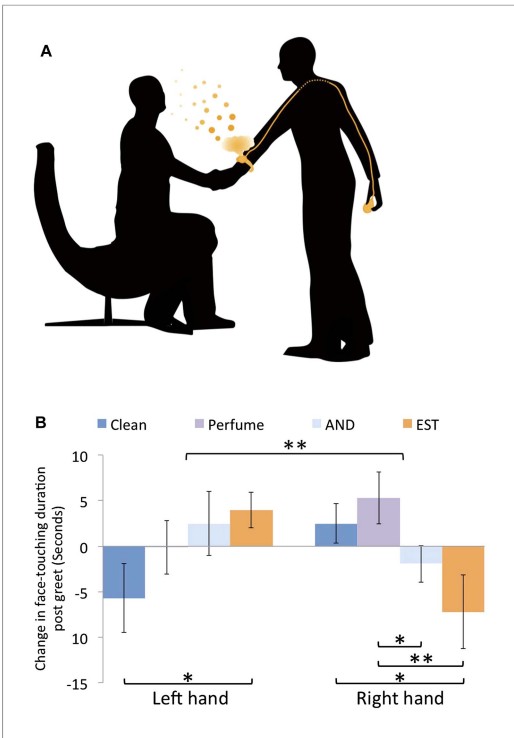

**Figure 4.** Tainting with odors alters post-handshake hand sniffing. (**A**) A schematic of the covert tainting device used for AND and EST. The non-shaking hand covertly activated a modified watch on the shaking hand that then emitted a plume of odor during handshake. (**B**) Face-touching behavior following tainting. Duration change scores are after individual-baseline normalization only. Error bars are standard error. **p < 0.01. *p < 0.05.

previous untainted 'within gender with hand-shake' condition in women (n = 22) (*Supplementary file 1*, Sheet 5). Note that although our covert AND/EST tainting device was wrist-worn (*Figure 4A*), it did not taint the shaking hand, and was designed to provide a general ambient subliminal body odor, much like wearing a perfume. In this analysis, each subject remains normalized to his/her own pre-greet baseline as before, but not further interaction-specific normalized (as in *Figure 3—figure supplement 1*). This is because, we did not collect a tainted no-handshake condition, which was here unnecessary because the critical comparison is of the same interaction, namely 'within gender with handshake' in women, just with taint or without. Our analysis addressed two hypotheses: chiefly, that odor would influence the behavior, and second that AND and EST would drive opposite effects in accordance with their hypothesized gender associations (*Savic et al., 2001*; *Zhou et al., 2014*). An ANOVA with factors of hand (L/R) and condition (AND/EST/CKB/clean) revealed no main effects, but a significant interaction of hand and condition (F[3,81] = 4.35, p < 0.007) (*Figure 4B*). Planned comparisons revealed that this reflected a decrease in self exploration of the shaking hand following tainting with the putative chemosignals compared to increased self exploration when no taint was used or after tainting with perfume (mean clean = 2.5 ± 10.1 s, mean EST = −7.2 ± 18.2, mean AND = −1.93 ± 9.5, mean perfume = 5.29 ± 13; clean vs EST: t[40] = 2.16, p = 0.04, perfume vs EST: t[39] = 2.53, p <

0.02, perfume vs AND: t[41] = 2.1, p = 0.04. Non-parametric reanalysis: Mann–Whitney U, clean vs EST Z = 1.97, p < 0.05, perfume vs EST Z = 2.2, p < 0.03). In contrast, a mirror image effect was evident in the non-shaking left hand. Here, there was an increase in self-exploration following tainting with putative chemosignals compared to a decrease when no taint was used and no change after tainting with perfume (mean clean = −5.7 ± 17.8 s, mean EST = 4 ± 8.8, mean AND = 2.47 ± 16.42, mean perfume = −0.12 ± 13.48; clean vs EST: t[40] = 2.2, p < 0.04. Non-parametric reanalysis: Mann–Whitney U, clean vs EST Z = 2.38, p < 0.02, clean vs AND Z = 2.36, p < 0.02) (Online *Video 3* final instance). In exit questionnaires administered following the EST and perfume controls, subjects were asked to provide a forced choice answer on whether an odor was present or not during the experiment. In the perfume condition, only 7 of 21 subjects noticed an odor (binomial cumulative P[X ≥ 7] = 0.96), and there was no difference in sampling behavior between those who did and did not notice an odor (Kolmogorov–Smirnov, X² = 2.38, p = 0.6). Similarly, in the EST condition, only 4 of 20 subjects noticed an odor (Binomial cumulative P[X ≥ 4] = 0.99), and here too there was no difference in sampling behavior between those who did and did not notice an odor (Kolmogorov–Smirnov, X² = 1.8, p = 0.81). Taken together, the first hypothesis materialized, and odors had a pronounced influence on the behavior of self-exploration after handshake. This influence remained consistent with the previously observed mirror behaviors of the left and right hands, and persisted despite lack of awareness for the odor manipulation. This strongly supports the subliminal olfactory nature of the behavior we measured. In contrast, the second hypothesis did not materialize, and there was no difference between AND and EST in this respect.

## Discussion

Whereas typical behavioral studies involve an ongoing task, here we observed subjects 'doing nothing' before and after a greeting event. A striking aspect of these observations was the extent of apparent olfactory self-exploration: human subjects repeatedly investigated their own hands, and this was often accompanied by overt sniffing (see online *Video 1* and *Video 2*). Our analyses reveal that such sniffing of the right shaking hand significantly increased selectively following handshakes with same gender individuals, and we speculate that this reflects chemo-investigation of conspecifics. Note that sniffing of the right shaking hand also persisted following cross-gender handshakes, yet here it dropped to a level significantly below baseline. Thus, this finding implies that humans are not only passively exposed to social chemosignals, but rather actively search for them. This was further evident in the tainting experiment where putative chemosignals and an ordinary perfume drove opposite effects. However, whereas previous studies found that AND and EST can bias perceptions in a gender-specific manner (*Savic et al., 2001*; *Zhou et al., 2014*), here AND and EST drove similar effects within gender that were opposite to that of a standard perfume. Whether the common response profile to AND and EST observed in the current setting has any bearing on their consideration as chemosignals is unlikely. Moreover, this issue is off-topic of the current study, where the important aspect is that one type of unnoticed odor taint decreased the behavior (AND and EST) yet a different type of unnoticed odor taint (perfume) did not. In other words, the behavior we measured was indeed influenced by unnoticed odor.

Our results were characterized by gender specificity that is common in social chemosignaling (*Doty, 1986*; *Savic et al., 2001*; *Bensafi et al., 2003*; *Dulac and Torello, 2003*; *Brennan and Zufall, 2006*; *Bergan et al., 2014*; *Kaur et al., 2014*). Although popular depictions of social chemosignaling typically highlight cross-gender interactions, a large number of documented social chemosignaling effects in both rodents and humans highlight the role of within gender social chemosignaling as well. For example, in rodents the chemosignal-mediated suppression of estrus (*Van Der Lee and Boot, 1955*) is a within gender effect. In humans, chemosignal-mediated menstrual synchrony (*Stern and McClintock, 1998*) is also a within gender effect. Therefore, the strong within gender effects observed in this study are not inconsistent with chemosignaling behavior. In turn, we speculate that the particular pattern we observed, namely increased sampling within gender, may be strongly modulated by context. In other words, we can imagine settings where one would perhaps increase investigation of the opposite gender rather than the same gender. Therefore, we think that the important aspects of our results are that people often sniff their own hands, and that they change this behavior after handshake. The specifics of the change, whether increase or decrease within or across gender, may be more specific to this study alone. Finally on this front, several studies imply that human chemosignaling is not only gender-specific, but also sexual-orientation-specific (*Savic et al., 2005*; *Berglund et al., 2006*; *Lubke et al., 2012*). We did not collect sexual orientation data, and therefore cannot say if the current within gender increase is strictly gender-specific, or perhaps also related to sexual orientation. This joins several unknowns regarding our result. For example, does familiarity between individuals influence this behavior? Might the behavior significantly shift across contexts? Is this behavior compensated for in some way in cultures where handshake is not common? These and more remain open questions for continued investigation.

The mechanism we propose serves to bridge the apparent gap between a role for social chemosignaling in ongoing human behavior and the lack of overt conspecific chemo-investigation. As noted in the introduction, beyond menstrual synchrony human chemosignals serve in mate selection (*Jacob et al., 2002*), convey fear (*Chen et al., 2006*; *Zhou and Chen, 2009*), drive pronounced hormonal (*Preti et al., 2003*; *Wyart et al., 2007*; *Gelstein et al., 2011*) and behavioral (*Jacob et al., 2001a*; *Bensafi et al., 2003*) modifications, and alter brain activity (*Sobel et al., 1999*; *Jacob et al., 2001b*; *Savic et al., 2001, 2005*; *Lundström et al., 2006*). Given these effects, which we speculate are only the tip of the iceberg, humans likely evolved social chemosignal-sampling strategies (*Arzi et al., 2014*), and we propose that handshaking is one of them. That said, we are not suggesting that social chemosignaling is the sole functional aspect of handshaking. Handshake orchestration conveys assorted social information (*Firth, 1972*; *Schiffrin, 1974*), contained within shake duration, posture, and strength (*Chaplin et al., 2000*). We do argue, however, that social chemosignaling may be a functional antecedent of handshaking and that it remains a meaningful albeit subliminal component of handshaking behavior.

Exploration of the right shaking hand was selectively increased following handshakes within gender. In turn, exploration of the left hand was ongoing, far more pronounced at rest, and selectively increased following handshakes with the opposite gender. We speculate that this reflects an ongoing comparative process whereby sniffing of the left hand subserves self-recognition and sniffing of the right hand subserves the investigation of others. This notion of a comparative process, however, remains a speculation deserving further address. Therefore, we conclude with reiterating the major findings of this study: first and foremost, humans apparently often sniff their own hands. Moreover, in the current context, after within gender handshakes humans significantly increase investigation of the hand that shook. This investigation is concurrent with pronounced sniffing, slightly increased after tainting the greeting experimenter with a perfume, yet is negated after tainting the greeting experimenter with putative social chemosignals. This combination leads us to conclude that handshaking may subserve sampling of social chemosignals. In addition to providing a functional framework for a common human behavior, these results imply an extensive role for social chemosignaling, which persists mostly without awareness for the signaling process.

## Materials and methods

All 281 subjects signed informed consent to procedures approved by the Wolfson Hospital Helsinki Committee. Moreover, after each experiment, subjects were offered the right to destroy the photographic data, or in turn provide specific consent for its use in research and/or publication. All subjects that appear in the accompanying videos provided written informed consent to have their video shown in scientific publications. Moreover, given the possibility of off-site reproduction by others, we obscured the facial features of subjects who consented to have their video shown in scientific publication but did not explicitly consent to have their video shown in non-scientific media. All experiments were conducted in stainless-steel coated odorant non-adherent rooms subserved by high throughput HEPA and carbon filtration that were specially designed for human olfaction experiments. For the initial measurement of possible chemosignal transfer by handshake (*Figure 1*) we measured three handshakes, one on each of three consecutive days, by each of the 10 participants (i.e., total 30 handshakes). Because we wanted to measure near-natural conditions, we did not instruct subjects to wash their hands before measurement. Instead, we collected data on all use of cosmetics. One subject (F) used hand cream and one subject (M) used facial cream. Other than one brand of hand-soap used by several subjects, there was no overlap in use of any cosmetic across the 10 subjects. We therefore conducted GCMS analysis of this soap, and did not find traces of any of the three components that occurred consistently across subjects. Thus, any of these components that transfer from handshake alone in all subjects cannot be attributed to a cosmetic source.

For GCMS analysis, we employed PDMS-covered stir bars (Gerstel Twister), rolled over the surface of the nitrile rubber glove both before and after a bare-skin hand shook it. The Twister was desorbed in a Thermal desorption Unit (TDU, Gerstel GmbH, Germany), with a temperature ramp of 20°C–170°C at 60°C/min, with a 5 min hold at maximum temperature. The Helium desorption flow was set at 40 ml/min in PTV solvent vent (1.2 min) and splitless TDU mode. The Programmable Temperature Vaporization Injector (PTV, Gerstel CIS4) was kept at −20°C through the desorption for trapping and focusing the transferred analytes on a quartz wool liner. Transfer line between the TDU and PTV was kept at 200°C. PTV temperature gradient was set to 12°C/s up to 250°C. Hold time at maximum temperature was 10 min. GC run was carried on an Agilent 7890 GC attached to an Agilent 5890 EI-single quadropole MSD. Restek Rxi-XLB 30 m × 0.25 mm × 0.25 µm medium polarity phase column was used. Oven program was 40°C for 3 min, then 15°C/min to 280°C for 5 min. Helium constant flow was at 1.2 ml/min. MS acquisition was carried out in TIC scan mode, 40–400 AMU. All MS parameters were automatically tuned. All resulting chromatograms were integrated according to the same parameters, using Agilent Chemstation software integrator. Peaks were screened for those occurring differentially in the two conditions, blank Twister run was used to screen out artifacts. Peaks were identified using NIST08/Wiley09 combined spectra library, and some peaks were confirmed by retention times and spectra with commercially obtained standards (Lactic acid, Glycerol, Squalene).

For airflow recording (*Figure 2D*), we used a nasal cannula linked to a spirometer (ADInstruments ML141) and instrumentation amplifier (ADInstruments PowerLab 16SP) recording at 1 kHz (*Johnson et al., 2006*).

For tainting experimenters with AND and EST, we used a specially devised odor-emitting modified watch (*Figure 4A*) that contained 500 µl of 2 mM compound (obtained from Steraloids Inc. Newport,

RI USA) dissolved in propylene glycol. Note that this was the concentration in a pad within the device, such that the puff of air that passed through it likely resulted in far lower ambient concentrations. Standard perfume was applied to the wrist at the same location of the watch.

For statistical analyses we used analyses of variance (ANOVA) followed by t-tests. We clearly state here that the decision to score 1 min before and 1 min after the greeting event (which culminated at ~80 s given the added time of the greet event itself, see *Figure 2—figure supplement 1*), rather than some other time window, was an a priori decision and not the result of a fishing expedition for time windows. Because subjects who did not sample their hand both before and after the greet during this 1 min were scored at zero (0) change in duration, this rendered the data abnormally distributed (Skew = −2.46, Kurtosis = 11). Although an ANOVA is relatively insensitive to this at sample sizes such as these (*Lix et al., 1996*), we nevertheless repeated each of the critical tests using a non-parametric approach as well (Mann–Whitney U test). The non-parametric results were mostly in full agreement with the parametric results and are reported throughout the manuscript. Finally, for simplicity in presentation, we computed a change from condition-specific baseline for each subject. This step is sensitive to extreme values in the no-handshake condition baseline. Thus, we repeated the analysis after deleting the highest and lowest extreme in each baseline, and we again obtained the same results with a reduction in power from $F(1,77) = 37.79$, $p < 0.0001$ to now $F(1,77) = 24.93$, $p < 0.0001$. This reanalysis is presented in *Supplementary file 1*, sheets 6–8. Finally, if we avoid condition-specific correction altogether and conduct the entire analysis with an additional 'Nature of Greet' level in the ANOVAs (with handshake/without handshake), the main results replicated in full, albeit with a slight further reduction in power to $F(1,145) = 12.75$, $p < 0.0005$. This somewhat more complex to follow graph together with its associated statistics are presented in *Figure 3—figure supplement 1* and in *Supplementary file 2*. Thus, the same result panned out when using baseline correction (*Figure 3*), when using baseline correction without extreme values at baseline (*Supplementary file 1*, Sheets 6–8), when avoiding baseline correction (*Figure 3—figure supplement 1*), and critically, when reverting to non-parametric statistics (text throughout the 'Results' section).

## Acknowledgements

We thank all 20 lab-member hand-shakers, and Professors M Saltman and M Sobel for anthropological and historical background on handshaking. Initiation of this work was funded by grants from the James S McDonnel Foundation and the Israeli Center for Excellence in Cognitive Neuroscience, and its continuation by the United States Air Force Office of Scientific Research, Program on Trust and Influence.

## Additional information

### Funding

| Funder | Grant reference number | Author |
| --- | --- | --- |
| James S. McDonnell Foundation (JSMF) | | Noam Sobel |
| Israeli Centers for Research Excellence (I-Core) | Israeli Center of Excellence in Cognitive Science | Noam Sobel |
| Air Force Office of Scientific Research (AFOSR) | Program on Trust and Influence | Noam Sobel |

The funders had no role in study design, data collection and interpretation, or the decision to submit the work for publication.

### Author contributions

IF, OP, YE-S, AE, NE, IH, MS, AR, LS, AA, NS, Conception and design, Acquisition of data, Analysis and interpretation of data, Drafting or revising the article

### Author ORCIDs

Idan Frumin, http://orcid.org/0000-0002-6293-1586

### Ethics

Human subjects: All subjects signed informed consent to procedures approved by the Wolfson Hospital Helsinki Committee (WOMC-0028-13). Moreover, after each experiment subjects were

offered the right to destroy the photographic data, or in turn provide specific consent for its use in research and/or publication. All subjects that appear in the accompanying videos provided written informed consent to have their video shown in scientific publications. Moreover, given the possibility of off-site reproduction by others, we obscured the facial features of subjects who consented to have their video shown in scientific publication but did not explicitly consent to have their video shown in non-scientific media.

## Additional files

### Supplementary files

• Supplementary file 1. Data and analysis. This excel file contains the analysis scheme, starting with raw duration values, and all the way through derivation of the data figures. Sheet 1 contains the initial data and its baseline correction. Sheet 2 contains the data separated by condition. Sheet 3 contains correction for condition. Sheet 4 contains the final data derivation. Sheet 5 contains data of the tainting experiment. Sheets 6 through 8 are a repetition of sheets 2 through 4 after deleting extremes from the condition-specific baseline.

• Supplementary file 2. ANOVA table for analysis without condition-specific correction. This excel file contains the ANOVA table for *Figure 3—figure supplement 1*. It reveals that the significant interaction is maintained when we do not correct for condition-specific baseline, and instead add it as an additional level in the analysis (with/without handshake).

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
