## [Decision Letter]

Thank you for sending your work entitled “A social chemosignaling function for human handshaking” for consideration at *eLife*. Your article has been favorably evaluated by a Senior editor and four reviewers, one of whom is Peggy Mason, a member of our Board of Reviewing Editors, and another of whom, Gün R Semin, agreed to reveal his identity.

The Reviewing editor and the other reviewers discussed their comments before we reached this decision, and the Reviewing editor has assembled the following comments to help you prepare a revised submission.

The reviewers were enthusiastic about the question of chemo-communication through handshaking, noting the study's novelty. The results are interesting, evocative, and sure to stimulate further more mechanistic studies on this phenomenon. There are several weaknesses that need to be addressed in a revision:

1) Figure 1 and the associated data show a chemical analysis of skin volatiles transferred onto a nitrile glove after shaking an ungloved hand. This work is not well enough explored to tell the reader anything relevant. What is the origin of the chemicals that are transferred? Could they be of exogenous origin, e.g. lotion or perfume? These data should be omitted.

2) As the authors themselves bring up, the “tainting” experiment illustrated in Figure 4 is poorly controlled. A same-sex control with women/EST and men/AND is needed as is the control mentioned in which a non-sex hormonal odor is used to taint the handshake. These data should be omitted.

3) Several methodological details are needed:

a) Does handedness make a difference? For most people, the left hand is both the non-shaking hand and the non-dominant hand and the opposite is true of the right hand. Whether handedness or shaking/non-shaking is the critical factor might be determined if it was known if the subjects were left or right-handed.

b) The independent treatment of the left and right hand sampling behavior is likely not warranted as the likelihood of sampling with both hands at the same time is probably nearly zero. If this is the case, the behaviors of R and L hand sampling are interdependent. This issue needs to be addressed both conceptually and statistically.

c) It is not clear what cover story the subjects were told nor whether any number of the subjects figured out that the researchers are interested in olfactory signaling. In this light, were subjects asked to not wear lotion, perfume or deodorant? If so, how was this request explained.

4) The language should be toned down. For example, handshaking could subserve social chemosignaling, but without any chemosignaling assay it is too early to conclude that it does. As another example, the statement “that humans constantly bring their hands to their nose and sniff” is an overstatement, one that is used more than once. During baseline, the hand goes to the face for 17 out of 60s. This is not even a majority of the time.

5) The reviewers discussed the ethological validity of the experiment. Questions regarding the familiarity of the handshaker and handshakee, the role of extended conversation and so on are likely to prove fodder for future research. The authors are invited to acknowledge these unknowns as such if they so choose.

---

## [Author Response]

*The reviewers were enthusiastic about the question of chemo-communication through handshaking, noting the study's novelty. The results are interesting, evocative, and sure to stimulate further more mechanistic studies on this phenomenon. There are several weaknesses that need to be addressed in a revision*.

We would like to state that we concur with each and all of the concerns raised, and we have acted to improve the manuscript in the relevant areas. In general, the major weaknesses identified by the editor and referees related not to the main study, but rather to two control studies. Although the editor opened the door to us simply deleting these control studies, we opted to take the longer path of in fact bettering them along the lines of the concerns raised by the editor and referees, as we think this makes for a more powerful end result. Thus, this revision includes not only new analysis and wording based on referees’ comments, but also new data from 50 additional subjects studied for this revision.

*1)*
Figure 1
*and the associated data show a chemical analysis of skin volatiles transferred onto a nitrile glove after shaking an ungloved hand. This work is not well enough explored to tell the reader anything relevant. What is the origin of the chemicals that are transferred? Could they be of exogenous origin, e.g. lotion or perfume? These data should be omitted*.

We completely concur with the above. Figure 1 in the original manuscript contained data from only 2 subjects (one man and one women), and it was not sufficiently detailed. We omitted this data. In turn, we think that ascertaining that handshakes can transfer chemosignaling relevant molecules is important for our major claim. With this in mind, we now repeated this experiment, now in 10 subjects, each tested 3 times (i.e., total 30 handshakes). Moreover, we now collected and detailed the relevant information on background. Specifically, because we wanted to measure near-natural conditions, we did not instruct subjects to wash their hands before measurement. Instead, we collected data on all use of cosmetics. One subject (F) used hand cream and one subject (M) used facial cream. Other than one common type of hand-soap used by several subjects, there was no overlap in use of any cosmetic across the 10 subjects. We therefore conducted GCMS analysis of this soap, and did not find traces of any of the three components that occurred consistently across all 10 subjects. Thus, any component that transfers from handshake alone in all subjects cannot be attributed to a cosmetic source. Given that the molecules we identify have been identified before as produced on human skin, we conclude that these were likely endogenous skin-bound molecules. These results do not imply that these molecules are necessarily social chemosignaling components in humans, but they do demonstrate that the act of handshaking is sufficient to transfer molecules of the type that is relevant to mammalian social chemosignaling.

*2) As the authors themselves bring up, the “tainting” experiment illustrated in*
Figure 4
*is poorly controlled. A same-sex control with women/EST and men/AND is needed as is the control mentioned in which a non-sex hormonal odor is used to taint the handshake. These data should be omitted*.

Again, we concur that the tainting control as presented in the original manuscript was not balanced. In turn, we think that demonstrating that odours influence the behaviour we observed is important if not critical. With that in mind, we studies an additional 40 subjects in order to balance AND and EST. We concentrated on women alone, and indeed also added a non-sex hormonal odour as an additional control. The end result of this was that both AND and EST reduced the behaviour equally, yet a standard perfume actually increased the behaviour (significantly above AND and EST, and nearly significantly above odourless baseline). This significant interaction (F(3,81) = 4.35, p < 0.007) is a critical “nail” in this box, as it demonstrates that the effect is odour driven. In addition to these detailed statistics, we also added a subject from the tainting experiment to Video 3. As to lack of difference between AND and EST, this has implications as to the profile of these two compounds, but we did not delve into this, as this manuscript is about chemosampling behaviour, not about AND and EST. Thus, we simply highlight that one set of odours reduced the behaviour and a different odour increased it. Hence, this is odour-driven behaviour. Importantly, we also added a post experimental exit questionnaire, and this revealed that subjects were unaware of the odour tainting. All this has been added to the manuscript.

*3) Several methodological details are needed*:

*a) Does handedness make a difference? For most people, the left hand is both the non-shaking hand and the non-dominant hand and the opposite is true of the right hand. Whether handedness or shaking/non-shaking is the critical factor might be determined if it was known if the subjects were left or right-handed*.

This is indeed an important point. We collected handedness information from all subjects, and now also report the relevant analysis. Consistent with population distribution, 15 of the 153 subjects and two of the 20 experimenters were left-handed. This retained four left-handed subjects in the “with handshake within gender” condition, three of which increased investigation of the shaking hand (right) after handshake. This reflects a trend towards a stronger effect in left-handed subjects, but this difference is not statistically different from the remaining right-handed subjects (X2 = 2.12, p = 0.14). Thus, here too our design protected against influence of individual differences such as handedness, and we cannot say whether handedness impacts this behaviour. All this has now been added to the manuscript.

*b) The independent treatment of the left and right hand sampling behavior is likely not warranted as the likelihood of sampling with both hands at the same time is probably nearly zero. If this is the case, the behaviors of R and L hand sampling are interdependent. This issue needs to be addressed both conceptually and statistically*.

First, we should note that simultaneous sampling of both hands did occur. This was not frequent, but far from zero. For example: see time stamp 00:19 and 01:40 of Video 3.

Given that these are left and right hands of the same individuals, a repeated-measures within-subjects ANOVA appeared the most appropriate approach. That said, we should note that we also followed up all the critical tests with non-parametric tests as well, and these treated each hand independently. Thus, given both parametric and independent nonparametric tests for each hand alone, we think that we have addressed this issue.

*c) It is not clear what cover story the subjects were told nor whether any number of the subjects figured out that the researchers are interested in olfactory signaling. In this light, were subjects asked to not wear lotion, perfume or deodorant? If so, how was this request explained*.

The cover story told to all subjects except those in the airflow recording control was that they should wait for an experimenter in the room they were directed to (note that the person who met them at the lab door and walked them to this room was a person they did not see again in the experiment). They were told to sit and wait for the experimenter, although this was of course already the beginning of the experiment. The experimenter then entered after three minutes, and conducted the “greet”. In the greet text, the experimenter told them to just continue waiting while he/she is preparing the experimental apparatus in the other room. Subjects were not told anything beyond this. As to lotions, etc., the critical aspect is of course not what subjects had on, but what experimenters had on. These were without any cosmetics. This is now clarified in the text.

*4) The language should be toned down. For example, handshaking could subserve social chemosignaling, but without any chemosignaling assay it is too early to conclude that it does. As another example, the statement “that humans constantly bring their hands to their nose and sniff” is an overstatement, one that is used more than once. During baseline, the hand goes to the face for 17 out of 60s. This is not even a majority of the time*.

We concur, and we have toned down at these and other locations.

*5) The reviewers discussed the ethological validity of the experiment. Questions regarding the familiarity of the handshaker and handshakee, the role of extended conversation and so on are likely to prove fodder for future research. The authors are invited to acknowledge these unknowns as such if they so choose*.

In the Discussion we now note: “We did not collect sexual orientation data, and therefore cannot say if the current within gender increase is strictly gender-specific, or perhaps also related to sexual orientation. This joins several unknowns regarding our result. For example, does familiarity between individuals influence this behaviour? Might the behaviour significantly shift across contexts? Is this behaviour compensated for in some way in cultures where handshake is not common? These and more remain open questions for continued investigation.”